# Deciphering Perceptual Quality in Colored Point Cloud: Prioritizing Geometry or Texture Distortion?

## ABSTRACT

Point cloud contents represent one of the prevalent formats for 3D representations. Distortions introduced at various stages in the point cloud processing pipeline affect the visual quality, altering their geometric composition, texture information, or both. Understanding and quantifying the impact of the distortion domain on visual quality is vital to driving rate optimization and guiding post-processing steps to improve the overall quality of experience. In this paper, we propose a multi-task guided multi-modality no reference metric for measuring the quality of colored point clouds (M3-Unity), which utilizes 4 types of modalities across different attributes and dimensionalities to represent point clouds. An attention mechanism establishes inter/intra associations among 3D/2D patches, which can complement each other, yielding both local and global features, to fit the highly nonlinear property of the human vision system. A multi-task decoder involving distortion type classification selects the best association among 4 modalities based on the specific distortion type, aiding the regression task and enabling the in-depth analysis of the interplay between geometrical and textural distortions. Furthermore, our framework design and attention strategy enable us to measure the impact of individual attributes and their combinations, providing insights into how these associations contribute particularly in relation to distortion type. Extensive experimental results on four benchmark datasets consistently outperform the state-of-the-art metrics by a large margin. The code will be released.

## CCS CONCEPTS

• **Human-centered computing** → **Visualization design and evaluation methods**; • **Computing methodologies** → **Perception**.

## KEYWORDS

Point cloud, Objective quality assessment, Multi-modal, multi-task, Geometry and texture

## 1 INTRODUCTION

Point cloud is prevailing among the available 3D imaging formats nowadays [16]. It consists of points in 3D space representing a geometric object realistically with various attributes, such as color, reflectance, and more. However, from acquisition to compression,

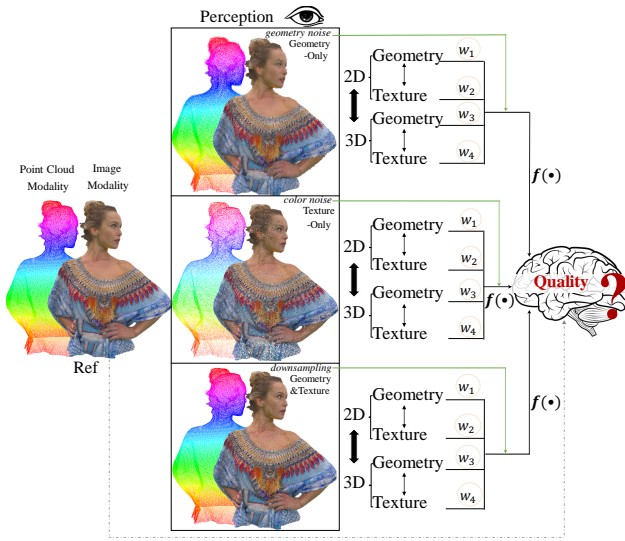

**Figure 1: A high level illustration of the proposed M3-Unity. The distortion can be divided into geometry and texture within the point cloud. The distortion type serves as a prior in shaping the perceptual quality through the HVS. The interplay of two attributes in representing entangled distortion adds complexity to this process.**

transmission, and rendering, the quality of a point cloud undergoes degradation. Consequently, there is a demand for effective and efficient objective Point Cloud Quality Assessment (PCQA) metric to guide the design, optimization, and parameter tuning of point cloud processing pipelines. PCQA metrics have been extensively utilized in various applications, including visual tasks such as restoration [10, 36], compression [22, 37], as well as for quality monitoring in various systems [9, 26, 38, 41].

Among all the visual artifacts for point clouds, the encountered distortions can generally be categorized into geometric and textural distortions, which can be created by compression algorithms as well as other noise-generation methods. Particularly in the context of lossy compression, approaches have been devised to encode geometric coordinates or associated attributes, depending on application requirements [33]. Given the necessity of color attributes for human visualization, it becomes imperative to combine algorithms addressing both geometric and textural to holistically represent the content. Therefore a large number of studies have been conducted to evaluate the quality of the point cloud in recent years subjectively and objectively [4]. Subjective studies investigate the quality of the point cloud under different distortion types of both geometry and texture attributes or of a single attribute [5]. Objective metrics also follow a similar paradigm to evaluate the quality. Early objective metrics primarily focused on geometric distortions, neglecting other

attributes. Geometric-based metrics, from a simple displacement such as point-to-point or point-to-plane [39] distance in the Euclidian space to a more complex geometric feature such as the point-to-distribution [18] and density-to-density [1] distances, examine the quality only from a geometric perspective. Color-based metrics [40, 43] produce a score computed only from color attributes. However, these metrics are hard to disentangle when distortions affect both attributes simultaneously, even when one attribute is not explicitly distorted (for example, distortion in geometry will affect the texture). Therefore the landscape has evolved to incorporate both geometry and texture [3, 7, 30, 51], with several approaches integrating multimodal learning for PCQA. MM-PCQA [56] first introduces multimodal learning for PCQA, combining uncolored point clouds and projected texture maps. MFT-PCQA [24] further improves the performance with a mediate-fusion strategy. pmBQA [47] perceives the quality by using four homogeneous modalities. Despite these advancements, existing metrics often overlook certain dimensionalities and fail to exploit the full potential of both attributes. Besides, the role of the distortion type is ignored. Furthermore, Lazzarotto *et al* [19] reveal that alternative trade-offs between geometry and texture can potentially provide better visual quality in a pair-wise comparison experiment. These researches shed certain light on how such interplay varies based on the distortion type as a first step towards this underexplored aspect in PCQA in a subjective manner. To the best of our knowledge, none of the existing metrics has explored how the geometric/textural distortion and their interplay contribute to the perceived quality of the point cloud automatically. Therefore, a more considerate design that can consider the interplay of such attributes in the Human Vision System (HVS) is needed.

Understanding these attributes and their interrelationships is crucial in various real-world applications. Nevertheless, the relative significance of each attribute representation as well as the interplay between them remain ambiguous in the context of PCQA, which reflects human perception preferences. As for which attribute is more important, we refer to specific distortion type. To this end, our metric, **M**ulti-**M**odality and **MU**lti-task **n**o reference quali**ty** assessment for colored point clouds, termed **M3-Unity**, investigate two attributes and their interplay for perceptual quality assessment. In particular, we use additional 3D normal and multi-view projections to retain the intrinsic characteristics of the point cloud and mimic the imaging process of HVS. Additionally, we measure the relationship between geometry and texture and their interplay given specific distortion type, as demonstrated in Figure 1. We evaluate the performance of the proposed metric on 4 independent datasets, i.e., SJTU-PCQA [49], WPC [21], Broad Quality Assessment of Static Point Clouds (BASICS) [2] and MJ-PCCD [19]. Our metric outperforms the state-of-the-art performance in terms of Pearson and Spearman correlation coefficients; moreover, the whole framework design elucidates the interplay between geometric and textural distortions, advancing the understanding of PCQA in real-world applications. To summarize, our key contributions are fourfold:

- We propose M3-Unity, a metric that uses 4 modalities across different attributes in different dimensionalities to represent the point cloud. The multi-task decoder involving distortion type classification selects the best combination among 4 modalities based on the distortion type, aiding in the regression task.

- The performance of M3-Unity and its variant demonstrates clear advantages over the state-of-the-art metrics across 4 datasets, showcasing substantial gains in comparison.
- We apply attention mechanism to establish inter/intra associations among patches (especially within dimensionality, we keep the spatial correspondence), which can complement each other, yielding both local and global features, to fit the highly nonlinear property of HVS.
- We delve into the relationship between geometric and textural distortion in terms of PCQA. Extensive experiments are conducted to determine whether geometric, textural, or their interplay is prioritized under various distortion types.

## 2 RELATED WORK

### 2.1 Subjective assessment of point clouds

Subjective quality assessment experiments are widely regarded as the most reliable method to evaluate the quality of point clouds, the interested reader may refer to [4] for a more detailed overview. Recently, many subjective studies have been conducted and reported in the literature to assess the performance of point cloud compression distortions in terms of visual quality. Lots of works present the subjective result for standard point cloud compression, such as base point cloud compression method from MPEG [29]; octree pruning using the Point Cloud Library and projection-based method implemented in the 3DTK toolkit [12]; Video-based Point Cloud Compression (VPCC) and Geometry-based PCC (GPCC) variants [6, 46]. Later, other distortion types are introduced in the SJTU-PCQA dataset [49] to mimic the acquisition and re-sampling noise besides the compression distortions. Liu *et al* [21] distorts the source point clouds with 4 processes to simulate real-world application scenarios and enrich the contents beyond those addressed by MPEG and JPEG. Liu *et al* [25] construct the largest dataset so far with pseudo-quality scores to support neural network training. 31 types of impairments covering a wide range of impairments during point cloud production, compression, transmission, and presentation are included. More recently, learning-based point cloud compression techniques have been considered. AK *et al* [2] include the GeoCNN compression distortion. Lazzarotto *et al* [19] first analyzes the impact of different configuration parameters on the performance of MPEG and JPEG Pleno compression with the aid of objective quality metrics.

### 2.2 Objective assessment of point clouds

Objective PCQA studies algorithms automatically evaluate the visual quality of point clouds as human judgments, it can be classified as Full-Reference (FR), Reduced-Reference (RR) and No-reference (NR) based on the availability of reference information. In this paper, we focus on deep-learning-based NR PCQA models.

PKT-PCQA [20] adopts a progressive knowledge transfer to convert the coarse-grained quality classification knowledge to the fine-grained quality prediction task. The key clusters are extracted based on global and local information, an attention mechanism is incorporated into the network design. Structure Guided Resampling (SGR) [58] considers that HVS is highly sensitive to structure information, it first exploits the unique normal vectors of point clouds to execute regional pre-processing. Then, three groups of quality-related features are extracted. Both the cognitive peculiarities of

the human brain and naturalness regularity are involved in the designed quality-aware features. These metrics belong to the single task with unimodal, which can not integrate the perception for both point cloud and image modality and is easy to overfit on the training data with only regression loss [45].

PQA-Net [23] takes 6 orthographic projections of point clouds as inputs, features are extracted after convolution neural network blocks, and they share a distortion identification and a quality prediction module to obtain final quality scores. GPA-net [34] proposes a graph convolution kernel, i.e., GPAConv, which attentively captures the perturbation of structure and texture, within a multi-task framework. A coordinate normalization module is utilized to stabilize the results of GPAConv under shift, scale and rotation transformations. PQA-Net [23] and GPA-Net [34] account for one main task (quality regression) and other auxiliary tasks (distortion type/degree predictions) when accessing only one modality of the point cloud.

IT-PCQA [50] utilizes the rich prior knowledge in images and builds a bridge between 2D and 3D perception in the field of quality assessment, a hierarchical feature encoder and a conditional discriminative network is proposed to extract latent features and minimize the domain discrepancy. pmBQA [47] proposes a projection-based blind quality indicator via multimodal learning by using four homogeneous modalities (i.e., texture, normal, depth and roughness). MM-PCQA [56] partitions point clouds into sub-models for local geometry representation and renders them into 2D projections for texture. A symmetric cross-modal attention module is used for integrating quality-aware information. IT-PCQA [50] reveals the potential connection between different types of media content for quality assessment. pmBQA [47] extract modality features by texture, normal, depth and roughness on 2D; MM-PCQA [56] proves the effectiveness of cross-modality perception for PCQA with texture on 2D and geometry on 3D. None of them considers the impact of distortion types. Remarkably, existing methods have not undertaken a comprehensive assessment that considers both dimensionality and attribute representations, while also incorporating multimodal within the framework of distortion types.

## 3 METHOD

We illustrate the proposed M3-Unity as shown in Figure 2. First, we preprocess the original colored point cloud data and extract multimodal features with 3D and 2D encoders, respectively (§3.2). Second, we introduce the cross-attributes attentive fusion module, which captures the local and global associations at both the intra- and inter-modality perception (§3.3). Last, we employ dual decoders to jointly learn both quality regression and distortion-type classification (§3.4).

### 3.1 Multimodal geometry-texture input processing

A colored point cloud, denoted as $\mathcal{P}$, is a set of $N$ 3D point elements. Each point element is assigned a 3D coordinate $\mathbf{p}^{\text{coord}} \in \mathbb{R}^3$ and an RGB color value $\mathbf{p}^{\text{RGB}} \in \mathbb{R}^3$ as features: $\mathcal{P} = \{(\mathbf{p}_i^{\text{coord}}, \mathbf{p}_i^{\text{RGB}})\}_{i=1}^N$. We introduce how the point cloud data is processed into multiple modalities of geometry and texture features as follows.

*Processing the point cloud as 3D patches.* To deal with dense point clouds of very large $N$ with common neural architectures for point cloud encoding, we first decompose each point cloud into patches following [44, 56]. we obtain a set of $n = 6$ point cloud patches from each of the original point cloud $\mathbf{P} \subset \mathcal{P}$, and each $\mathbf{P}$ is of cardinality $k$. To do this, we adopt Farthest Point Sampling (FPS) to obtain a set of anchor points and find the K-Nearest Neighbors (KNN) for each point. For each point cloud patch $\mathbf{P}$, we describe the geometry and texture features for each point element, such that the texture features are essentially the RGB features $\mathbf{p}^{\text{tex}} = \mathbf{p}^{\text{RGB}} \in \mathbb{R}^3$, and the geometry feature is the 3D coordinate $\mathbf{p}^{\text{coord}}$, augmented by concatenating a normal vector $\mathbf{p}^{\text{normal}}$ calculated from the original point cloud as $\mathbf{p}^{\text{geo}} = [\mathbf{p}^{\text{coord}}, \mathbf{p}^{\text{normal}}] \in \mathbb{R}^6$, i.e. $\mathbf{P} = \{(\mathbf{p}_i^{\text{geo}}, \mathbf{p}_i^{\text{tex}})\}_{i=1}^k$. Additionally, $\mathbf{P} \in \mathbb{P}$ where $\mathbb{P}$ is defined as the set of all 3D point patches extracted from the same point cloud.

*Processing the point cloud as projected views.* We further project the colored point cloud to $m = 6$ 2D views following Liu et al. [23], which are evenly distributed in the 3D space from the $\infty$ and $-\infty$ of the three Cartesian coordinate axes. For each 2D view, the color RGB values from the 3D points are ray-casted to the pixel space, and we calculate depth and normal maps from the 3D geometry, resulting in the 2D geometry feature $\mathbf{X}^{\text{geo}} \in \mathbb{R}^{H \times W \times 4}$ and the 2D texture feature $\mathbf{X}^{\text{tex}} \in \mathbb{R}^{H \times W \times 3}$, where $H \times W$ is the pixelated resolution of the 2D projections. Similarly we define $\mathbb{X}$ as the set of six projected views from a point cloud: $\mathbf{X} = [\mathbf{X}^{\text{geo}}, \mathbf{X}^{\text{tex}}] \in \mathbb{X}$.

### 3.2 Point cloud multimodal encoding

The goal of multimodal encoding is to represent 3D point cloud patches and 2D projection views as embeddings and adapt those embeddings for multimodal fusion.

For the 3D modality, we opt for PointNet++ [32] to encode each 3D point cloud patch $\mathbf{P} = \{(\mathbf{p}_i^{\text{geo}}, \mathbf{p}_i^{\text{tex}})\}_{i=1}^k \subset \mathbb{P}$ while separating attributes from geometry and texture:

$$\mathbf{h}_{3D}^{\text{geo}} = \text{PointNet++}\left(\{\mathbf{p}_i^{\text{geo}}\}_{i=1}^k\right); \tag{1}$$

$$\mathbf{h}_{3D}^{\text{tex}} = \text{PointNet++}\left(\{\mathbf{p}_i^{\text{tex}}\}_{i=1}^k\right). \tag{2}$$

$\mathbf{h}_{3D}^{\text{geo}} \in \mathbb{R}^d$ and $\mathbf{h}_{3D}^{\text{tex}} \in \mathbb{R}^d$ are $d$-dimensional embeddings of 3D geometry and texture features. Note that to encode texture feature, we still use the 3D coordinates to obtain spatial processes in the PointNet++ such as the farthest-point sampling and grouping.

For the 2D modality, we choose ResNet50 [17] as the 2D encoder that applies to the geometry and texture channels $\mathbf{X}^{\text{geo}}$ and $\mathbf{X}^{\text{tex}}$ separately of each 2D project view $\mathbf{X} \in \mathbb{X}$:

$$\mathbf{h}_{2D}^{\text{geo}} = \text{ResNet}\left(\mathbf{X}^{\text{geo}}\right); \tag{3}$$

$$\mathbf{h}_{2D}^{\text{tex}} = \text{ResNet}\left(\mathbf{X}^{\text{tex}}\right). \tag{4}$$

Likewise, $\mathbf{h}_{2D}^{\text{geo}} \in \mathbb{R}^d$ and $\mathbf{h}_{2D}^{\text{tex}} \in \mathbb{R}^d$ are encoded as $d$-dimensional 2D geometry and texture embeddings.

### 3.3 Cross-attribute attentive fusion

The core mechanism of attention gains popularity for capturing the associations when processing images [11, 28, 44, 53]. We employ patch attention [15, 54] to capture the local and global associations for both intra- and inter-modality features, followed by a symmetric

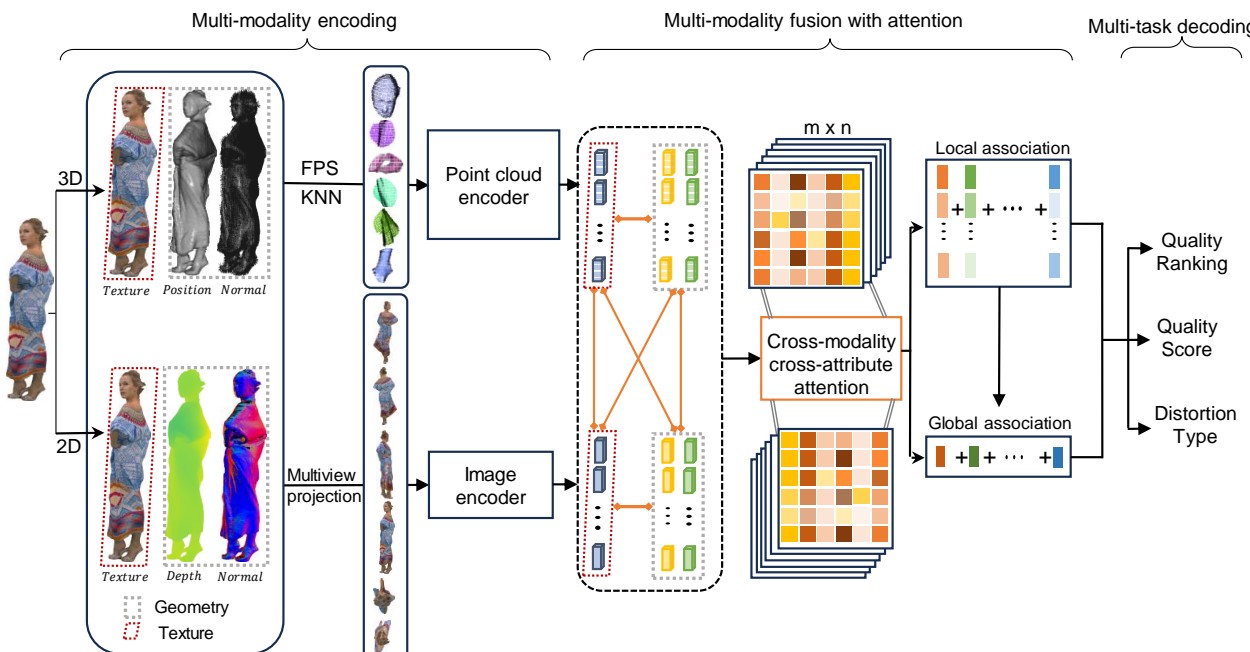

**Figure 2: M3-Unity architecture: no-reference multi-modality and multi-task learning for PCQA.**

fusion function that averages the cross-attented features to model the symmetric interaction of the source pair of features.

*Symmetric intra-modality attentions.* For each 3D point cloud patch $\mathbf{P} \in \mathbb{P}$, we employ intra-modality attention by applying the symmetric fusion function $\Psi^*(\cdot, \cdot)$ to encode the interrelationship of geometry and texture features. For simpler notation, we assign a random sequence for the patches and arrange the set of the features extracted features $\mathbf{h}_{3D}^{geo}$ and $\mathbf{h}_{3D}^{tex}$ for all patches in forms of matrices as $\mathbf{H}_{3D}^{geo} \in \mathbb{R}^{n \times d}$ and $\mathbf{H}_{3D}^{tex} \in \mathbb{R}^{n \times d}$.

The 3D intra-modality attentive fusion becomes

$$\mathbf{H}_{3D}^{intra} = \Psi^*(\mathbf{H}_{3D}^{geo}, \mathbf{H}_{3D}^{tex}) \in \mathbb{R}^{n \times d}. \tag{5}$$

$$\mathbf{h}_{3D}^{intra} = \text{Mean}(\mathbf{H}_{3D}^{intra}) \in \mathbb{R}^{d}, \tag{6}$$

where $\text{Mean}(\cdot)$ is the mean pooling over the sequence dimension to achieve the global feature for the entire point cloud from aggregating all patches in an attentive manner. $\Psi^*(\cdot, \cdot)$ is the symmetric fusion function based on the attention function $\Psi(\cdot, \cdot)$ such that:

$$\Psi^*(\mathbf{x}, \tilde{\mathbf{x}}) = \frac{1}{2} \left( \Psi(\mathbf{x}, \tilde{\mathbf{x}}) + \Psi(\tilde{\mathbf{x}}, \mathbf{x}) \right), \tag{7}$$

which assumes equal sequence dimensions $l_1 = l_2$ of the Query and Key in the transformer. And $\Psi(\cdot, \cdot)$ is the basic fusion transformer, which is computed by an attentive representation of a target modality referred to a reference modality in the multi-head self-attention.

Similarly for the 2D modality $\mathbb{X}$, we define $\mathbf{H}_{2D}^{geo} \in \mathbb{R}^{m \times d}$ and $\mathbf{H}_{2D}^{tex} \in \mathbb{R}^{m \times d}$, and the 2D intra-modality attention is

$$\mathbf{h}_{2D}^{intra} = \text{Mean}(\mathbf{H}_{2D}^{intra}) = \text{Mean}(\Psi^*(\mathbf{H}_{2D}^{geo}, \mathbf{H}_{2D}^{tex})) \in \mathbb{R}^{d}. \tag{8}$$

We clarify that the random sequence assignment would not affect the final output feature detailed as follows, since the attention function is equivariant to the permutation of the sequence, and we

will average over the sequence dimension to aggregated feature output.

*Symmetric inter-modality attention.* For inter-modality attentive features, we cross-attend each pair of 3D point cloud patch and 2D projection view in the combinatorial set $\{\mathbf{P}, \mathbf{X}\} \in \mathbb{P} \times \mathbb{X}$. We employ the inter-modality attention by applying $\Psi^*(\cdot, \cdot)$ across 3D and 2D modalities. Note that this result can only be achieved when we have the same number of 3D patches and 2D projections $n = m$ for each point cloud. In the rest of this section, we will discard the notation of $m$ and consistently use $n$ for $|\mathbb{P}| = |\mathbb{X}| = 6$ to reduce confusion.

$$\begin{aligned}
\mathbf{H}_{inter}^{geo\text{-}geo} &= \Psi^*(\mathbf{H}_{3D}^{geo}, \mathbf{H}_{2D}^{geo}) \in \mathbb{R}^{n \times d} \\
\mathbf{H}_{inter}^{geo\text{-}tex} &= \Psi^*(\mathbf{H}_{3D}^{geo}, \mathbf{H}_{2D}^{tex}) \in \mathbb{R}^{n \times d} \\
\mathbf{H}_{inter}^{tex\text{-}geo} &= \Psi^*(\mathbf{H}_{3D}^{tex}, \mathbf{H}_{2D}^{geo}) \in \mathbb{R}^{n \times d} \\
\mathbf{H}_{inter}^{tex\text{-}tex} &= \Psi^*(\mathbf{H}_{3D}^{tex}, \mathbf{H}_{2D}^{tex}) \in \mathbb{R}^{n \times d}.
\end{aligned} \tag{9}$$

Similar to Eq. 6, we apply average pooling $\text{Mean}(\cdot)$ to obtain global inter-modality attentive features $\mathbf{h}_{inter}^{geo\text{-}geo}$, $\mathbf{h}_{inter}^{geo\text{-}tex}$, $\mathbf{h}_{inter}^{tex\text{-}geo}$, and $\mathbf{h}_{inter}^{tex\text{-}tex}$ for the entire point cloud.

*Feature aggregation.* We aggregate all multi-modal geometry and texture features as well as all intra- and inter-modality attentive features for the final feature encoding:

$$\begin{aligned}
\mathbf{h} = &\mathop{\mathbb{E}}_{\mathbf{P}_i \in \mathbb{P}} [\mathbf{h}_{3D,i}^{geo} + \mathbf{h}_{3D,i}^{tex}] + \mathop{\mathbb{E}}_{\mathbf{X}_j \in \mathbb{X}} [\mathbf{h}_{2D,j}^{geo} + \mathbf{h}_{2D,j}^{tex}] \\
&+ \frac{\mathbf{h}_{3D}^{intra} + \mathbf{h}_{2D}^{intra}}{2} + \frac{\mathbf{h}_{inter}^{geo\text{-}geo} + \mathbf{h}_{inter}^{geo\text{-}tex} + \mathbf{h}_{inter}^{tex\text{-}geo} + \mathbf{h}_{inter}^{tex\text{-}tex}}{4}.
\end{aligned} \tag{10}$$

The resulting feature $\mathbf{h}$ serves as the input to the decoder heads for final predictions, to be detailed as follows.

## 3.4 Multi-task learning with dual decoders

*Dual decoders.* We define dual decoders using multi-layer perception for quality regression and distortion-type classification respectively with a regression head $\psi_{\text{regression}}$ and a classification head $\psi_{\text{classification}}$, both taking the aggregated feature $\mathbf{h}$ as the input. The regression head $\psi_{\text{regression}}$ is a two-layer ReLU-MLP that outputs $y$ the quality score:

$$y = \psi_{\text{regression}}(\mathbf{h}) \in \mathbb{R}. \tag{11}$$

The classification head $\psi_{\text{classification}}$ is a three-layer ReLU-MLP with a softmax activation attached to the output layer, which gives $z$ the one-hot prediction of classification type:

$$z = \psi_{\text{classification}}(\mathbf{h}) \in \mathbb{R}^c, \tag{12}$$

where $c$ is the number of types of distortions.

*Learning loss.* We define and jointly learn the dual decoders by a triplet learning loss $\mathcal{L}$ for a mini-batch with size of $n$ as:

$$\mathcal{L} = \lambda_1 \mathcal{L}_{\text{mse}} + \lambda_2 \mathcal{L}_{\text{rank}} + \lambda_3 \mathcal{L}_{\text{ce}}, \tag{13}$$

where $\lambda_1, \lambda_2, \lambda_3 \in [0, 1]$ are importance scores used to control the proportion of each type of loss.

Specifically, we compute Mean Square Error (MSE) loss between predicted quality scores and human scores as:

$$\mathcal{L}_{\text{mse}} = \frac{1}{n} \sum_{i=1}^{n} (y_i - y_i')^2. \tag{14}$$

We compute ranking loss of the predicted quality scores and human scores as:

$$\mathcal{L}_{\text{rank}} = \frac{1}{n^2} \sum_{i=1}^{n} \sum_{j=1}^{n} l_{ij}, \quad \text{where}$$

$$l_{ij} = \max\left(0, \left|y_i - y_j\right| - (-1)^{\mathbb{1}(y_i < y_j)} \cdot \left(y_i' - y_j'\right)\right). \tag{15}$$

Here $i$ and $j$ are the corresponding indexes for two point clouds in a mini-batch, and $\mathbb{1}(\cdot)$ is the indicator function.

We compute the cross-entropy loss of the predicted distortion type and the ground-truth labels:

$$\mathcal{L}_{\text{ce}} = \frac{1}{n} \sum_{i=1}^{n} -\left(z_i' \log(z_i) + (1 - z_i') \log(1 - z_i)\right) \tag{16}$$

## 4 EXPERIMENTAL SETUP

*Datasets.* We employ the SJTU-PCQA [49], WPC [21], BASICS [2] and MJ-PCCD [19] datasets for validation. SJTU-PCQA includes 9 reference point clouds and each one is corrupted with 7 distortion types (octree-based compression, color noise, downscaling, downscaling & color noise, downscaling & geometry Gaussian noise, geometry Gaussian noise, color noise & geometry Gaussian noise), which generates 378 distorted stimuli. WPC contains 20 reference point clouds with each one degraded under 5 distortion types (VPCC, Gaussian noise, downsampling, GPCC (Octree/Trisoup)), leading to 740 distorted stimuli. BASICS comprises 75 point clouds from 3 different semantic categories. Each point cloud is compressed with 4 different algorithms (geoCNN, GPCC-octree-RAHT, GPCC-octree-Predlift, VPCC) at varying compression levels, resulting in 1494 processed stimuli. MJ-PCCD was created by compressing 6 point clouds from the JPEG Pleno test set at 4 different bitrates with

the GPCC, VPCC, and JPEG Pleno standards, producing a total of 213 distorted stimuli.

*Evaluation Criteria.* Three commonly used evaluation criteria are used to reflect the relationship between objective scores and subjective scores: (1) Pearson Linear Correlation Coefficient (PLCC), which measures the linearity of prediction; (2) Spearman Rank-order Correlation Coefficient (SRCC), which measures the monotonicity of prediction; (3) Root MSE (RMSE), which measures the error of prediction. Higher values of PLCC and SRCC indicate better performance in terms of correlation with human opinion, while lower RMSE indicates better consistency. A five-parametric logistic regression is additionally adopted to fit the relationship between the subjective scores and the objective scores [8]

*Comparable methods.* We selected 13 state-of-the-art PCQA metrics for comparison, which consist of 5 FR metrics, 2 RR metrics and 6 learning-based NR metrics. The FR metrics include PCQM [30], GraphSIM [51], PointSSIM [3], MPED [52] and PointPCA [7]. The RR metric are PCM-RR [42] and RR-CAP [59]. The NR metrics include 3D-NSS [55], IT-PCQA [50], VS-ResNet [14], MM-PCQA [56], GMS-3DQA [57].

*Implementation details.* The proposed M3-Unity is implemented based on the PyTorch [31]. We use Adam optimizer [27] with weight decay as 1e-4, the initial learning rate is set as 5e-5, and the batch size is set as 4. The model is trained for 100 epochs. Specifically, we set the cardinality $k$ of a local point cloud patch as 2048, the number of local patches and the number of image projections both equal to 6. The projected images with the resolution of 1920×1080, the cropped image patches at the resolution of 224×224. We use Point-Net++ [32] as the point cloud encoder and initialize ResNet50 [17] with a pre-trained model on ImageNet [13]as the image encoder. The multi-head attention module employs 8 heads and the feed-forward dimension is 2048. The MOS values among the datasets are scaled between [1, 10]. $\lambda_1$, $\lambda_2$ and $\lambda_3$ are all equal to 1. Following the practices in [23], We employ a k-fold cross-validation strategy to evaluate the performance of the proposed method accurately. We conduct 9/5/6-fold cross-validation for SJTU-PCQA, WPC and MJ-PCCD datasets and report average scores as the final performances. For the BASICS dataset, we follow the 60%-20%-20% training-validation-testing strategy and compute the performance on the test set. Notably, there is no content overlap between training and testing sets. For FR PCQA metrics requiring no training, we assess them on the same testing sets.

## 5 RESULTS

### 5.1 Overall Performance

Results of SRCC and PLCC on 4 datasets for the proposed M3-Unity and other 13 PCQA metrics are shown in Table 1. First, the proposed M3-Unity significantly outperforms the compared metrics in terms of SRCC on all datasets. Second, compared with GMS-3DQA, which uses the projection-based grid mini-patch sampling only from image modality, the PLCC decreases by 0.017 on the MJ-PCCD. One possible reason is there are super dense/sparse point clouds in MJ-PCCD. Therefore, the projection takes effect when revealing the overlap/hole. While compared with MM-PCQA, which uses 2 modalities from 3D and 2D, M3-Unity is better across 4 datasets,

Table 1: Performance comparison among the proposed and the state-of-the-art PCQA metrics on SJTU-PCQA, WPC, BASICS, and MJ-PCCD datasets. Best in bold and second with underlined fonts. Please note that the state-of-the-art results were taken from the literature, often with different training strategies and splits, and not independently validated by the authors.

| Category | Method | SJTU-PCQA | | WPC | | BASICS | | MJ-PCCD | |
|---|---|---|---|---|---|---|---|---|---|
| | | SRCC | PLCC | SRCC | PLCC | SRCC | PLCC | SRCC | PLCC |
| FR | PointSSIM [3] | 0.687 | 0.714 | 0.454 | 0.467 | 0.692 | 0.725 | 0.467 | 0.597 |
| | PCQM [30] | 0.864 | 0.885 | 0.743 | 0.750 | 0.810 | 0.888 | 0.779 | 0.858 |
| | GrahSim [51] | 0.878 | 0.845 | 0.583 | 0.616 | / | / | 0.758 | 0.844 |
| | MPED [52] | 0.898 | 0.915 | 0.620 | 0.618 | 0.761 | 0.835 | 0.735 | 0.811 |
| | PointPCA [7] | 0.907 | 0.932 | 0.890 | 0.894 | 0.866 | 0.926 | 0.834 | 0.702 |
| RR | PCM-RR [42] | 0.482 | 0.336 | 0.310 | 0.343 | 0.436 | 0.518 | 0.497 | 0.636 |
| | RR-CAP [59] | 0.758 | 0.769 | 0.716 | 0.731 | 0.558 | 0.740 | 0.550 | 0.735 |
| NR | IT-PCQA [50] | 0.630 | 0.580 | 0.568 | 0.561 | 0.310 | 0.302 | 0.658 | 0.807 |
| | 3D-NSS [55] | 0.714 | 0.738 | 0.648 | 0.651 | 0.617 | 0.657 | 0.446 | 0.411 |
| | ResSCNN [25] | 0.810 | 0.860 | 0.735 | 0.752 | 0.628 | 0.682 | 0.759 | 0.842 |
| | VS-ResNet [14] | 0.830 | 0.860 | 0.760 | 0.770 | 0.711 | 0.852 | 0.526 | 0.583 |
| | MM-PCQA [56] | 0.910 | 0.923 | 0.841 | 0.856 | 0.831 | 0.882 | 0.860 | 0.898 |
| | GMS-3DQA [57] | 0.911 | 0.918 | 0.831 | 0.834 | 0.855 | 0.930 | 0.879 | **0.936** |
| | M3-Unity(Proposed) | **0.947** | **0.961** | **0.900** | **0.900** | **0.872** | **0.937** | **0.903** | 0.919 |

Table 2: Cross-dataset validation among SJTU-PCQA, WPC, BASICS and MJ-PCCD datasets. Both the training and testing are on the complete dataset.

| | Test | | | | | | | | | | | |
|---|---|---|---|---|---|---|---|---|---|---|---|---|
| | SJTU-PCQA | | | WPC | | | BASICS | | | MJ-PCCD | | |
| Train | SRCC | PLCC | RMSE | SRCC | PLCC | RMSE | SRCC | PLCC | RMSE | SRCC | PLCC | RMSE |
| SJTU-PCQA | – | – | – | 0.444 | 0.473 | 2.020 | 0.537 | 0.671 | 0.794 | 0.457 | 0.701 | 0.835 |
| WPC | 0.821 | 0.841 | 1.314 | – | – | – | 0.617 | 0.712 | 0.752 | 0.643 | 0.767 | 0.751 |
| BASICS | 0.523 | 0.559 | 2.013 | 0.509 | 0.514 | 1.967 | – | – | – | 0.825 | 0.867 | 0.582 |
| MJ-PCCD | 0.635 | 0.653 | 1.838 | 0.440 | 0.507 | 1.976 | 0.779 | 0.827 | 0.602 | – | – | – |

that's because we utilized multi-attributes for both dimensionalities and the interplay among them. In summary, M3-Unity demonstrates robust and competitive performance across 4 benchmarks. This validates our motivation that incorporating multi-attributes in both dimensionalities and the interplay contributes to improved perceptual quality inference.

## 5.2 Cross Dataset Validation

To verify the generalization and robustness of the proposed M3-Unity, we conduct cross-dataset experiments among all datasets. The experimental results are shown in Table 2. From Table 2, we can see that M3-Unity has good generalization performance, the cross-dataset performance is even higher than certain FR PCQA metrics, for example, the performance is higher than PointSSIM when training on WPC and testing on SJTU-PCQA (the SRCC of MM-PCQA [56] is 0.769, and the PLCC of CoPA [35] is 0.643) and MJ-PCCD datasets.

## 5.3 Time and complexity analysis

M3-unity contains 89.1M parameters using around 37GB GPU memory with batch size 4 and has an average inference time of 0.55 seconds for 1 PC from the SJTU-PCQA dataset on A100.

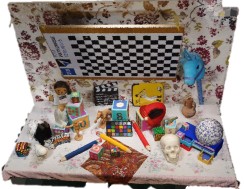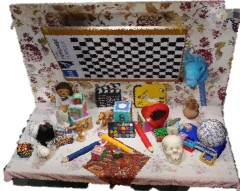

MOS: 9.117

MOS: 4.591
Geometry-Only: 5.642    PointSSIM: 4.327
Texture-Only:   4.734    Y_PNSR:   4.445

Figure 3: Exemplified point cloud *Unicorn* comparing predictions between learning-based metrics and traditional FR metrics. The left side is the reference version of *Unicorn*, and the right side is the distorted version of it. The distortion is geometry Gaussian noise: all the points are augmented with a random geometric shift within 0.02%.

## 5.4 Ablation Study

We conduct an ablation study on M3-Unity to examine the impact of key components for the performance. Additionally, in the context of the 4 datasets characterized by distinct content and distortion types, we categorized each dataset into Human and Animal (HA) and Inanimate Object (IO) subsets and reported the related performance. Note: WPC only includes IO.

Table 3: Ablation study of M3-Unity on key components, i.e., distortion type, attention, and modality. The numbers in brackets denote the performance of the IB and HA, with the best performance highlighted in blue and orange, respectively.

| Settings | SJTU-PCQA | | | WPC | | | BASICS | | | MJ-PCCD | | |
|---|---|---|---|---|---|---|---|---|---|---|---|---|
| | SRCC | PLCC | ACC | SRCC | PLCC | ACC | SRCC | PLCC | ACC | SRCC | PLCC | ACC |
| M3-Unity | **0.947** | **0.961** | 0.728 | 0.900 | 0.900 | 0.981 | 0.872 | **0.937** | **0.847** | **0.903** | 0.919 | **0.643** |
| | (**0.933**\|0.964) | (**0.949**\|0.964) | (**0.583**\|0.795) | / | / | / | (0.867\|0.889) | (0.925\|0.929) | (0.810\|0.840) | (0.858\|**0.892**) | (0.908\|0.905) | (0.545\|0.552) |
| *(Distortion Type)* | | | | | | | | | | | | |
| /wo DT classification | 0.938 | 0.951 | | 0.898 | 0.898 | | 0.856 | 0.924 | | 0.900 | 0.924 | |
| | (0.930\|0.963) | (0.948\|0.966) | / | / | / | / | (0.860\|0.872) | (0.916\|0.933) | / | (**0.873**\|0.883) | (**0.917**\|0.920) | / |
| *(Attention)* | | | | | | | | | | | | |
| /wo patch attention | 0.919 | 0.941 | 0.537 | 0.849 | 0.855 | 0.969 | 0.684 | 0.733 | 0.730 | 0.846 | 0.869 | 0.590 |
| | (0.876\|0.946) | (0.921\|0.950) | (0.446\|0.671) | / | / | / | (0.691\|0.777) | (0.802\|0.807) | (0.525\|0.740) | (0.808\|0.853) | (0.847\|0.881) | (**0.611**\|0.587) |
| *(Modality)* | | | | | | | | | | | | |
| /wo 2D projection | 0.914 | 0.947 | 0.595 | 0.608 | 0.638 | 0.792 | 0.770 | 0.638 | 0.610 | 0.736 | 0.812 | 0.492 |
| | (0.886\|0.947) | (0.938\|0.954) | (0.542\|0.610) | / | / | / | (0.759\|0.771) | (0.850\|0.815) | (0.565\|0.650) | (0.533\|0.776) | (0.664\|0.838) | (0.462\|0.403) |
| /wo 3D point cloud | 0.943 | 0.957 | **0.773** | **0.911** | **0.912** | **0.989** | 0.879 | 0.937 | 0.843 | 0.896 | **0.931** | 0.624 |
| | (0.900\|**0.967**) | (0.941\|**0.971**) | (0.571\|**0.795**) | / | / | / | (**0.872**\|**0.890**) | (**0.930**\|**0.945**) | (**0.905**\|**0.880**) | (0.860\|0.880) | (0.912\|**0.936**) | (0.575\|**0.636**) |

*Impacts of distortion type classification.* To verify the effect of the distortion type classification module, we compare the performance with only the regression decoder. The result is in Table 3 (Distortion Type). There is a slight performance drop for the 4 datasets when the distortion type classification task is omitted. Significantly, the prediction accuracy (ACC) of distortion types differs considerably between the WPC and MJ-PCCD datasets. ACC represents the proportion of correct predictions out of the total. There is no discernible direct correlation between the accuracy of distortion type classification and the accuracy of quality prediction with the current datasets.

*Impacts of the modalities.* Intuitively, combining 4 modalities can gain better visual representations than unimodal. The performance comparison of the 4 datasets is in Table 3. The performance of both uni-modal-based models is inferior or quite similar to M3-Unity except for the WPC dataset, which suggests that both modalities make contributions to creating perceptual representations. What's more, image-based modality is more important than point cloud-based since HVS takes pictures of point clouds first and then processes the visual stimuli.

*Impacts of the attention.* The self-attention mechanism calculates semantic affinities between different items in a sequence of data [15], i.e., we capture the local context within the point cloud, by enhancing input embedding with the support of FPS and KNN search. Upon removing the attention module, the results are presented in Table 3 (Attention). M3-Unity exhibits superiority in comparison to the model without the attention mechanism.

Our investigation revealed that M3-Unity and its variants consistently demonstrate superior performance on HA compared to IO data, as measured by SRCC across all datasets, with the number of HA being more than or equal to that of IO for SJTU-PCQA and MJ-PCCD datasets. Specifically, we observed that patch attention predominantly influences performance for the SJTU and BASICS datasets, whereas 2D projection assumes a pivotal role for the WPC and MJ-PCCD datasets within the framework of M3-Unity, relative to other components. Upon further analysis, we found that excluding the patch attention component resulted in a performance drop of 9.4% for IO data and 6.2% for HA data. Similarly, when excluding the 2D projection component, the performance drop was more pronounced, with reductions of 21.8% for IO data and 9.3% for HA data. Remarkably, IO data consistently exhibited a greater decline in performance compared to HA data across the datasets, except for the BASICS dataset, where the performance decrement was comparable for both categories.

## 5.5 Discussion

We examine the interplay of geometry and texture distortion representations in composite distortions and explore their associations across dimensionalities.

*Interplay between geometry and texture.* To further explore which distortion representation is allocated more attention when encountering various degradations, we predict the quality with geometry-only (3D position, normal point clouds, 2D depth, normal maps) and texture-only (3D texture point cloud, 2D texture map) features, separately. The performance on the 4 datasets is in Table 4.

In addition, we assessed the quality of the distorted point cloud by examining it from both geometry-only and texture-only perspectives in comparison to the reference one. Figure 3 illustrates the results obtained by the variants of M3-Unity alongside the results from FR PCQA metrics. Specifically, we use the average of norm and curvature of PointSSIM [3] as the geometry measurement, while Y_PNSR serves as the texture measurement. In the FR manner, Y_PNSR exhibits greater similarity to the referenced MOS (9.117) than geometry, underscoring the predominant role of texture-related representation in predicting the quality of the *Unicorn* point cloud. Notably, our model's prediction (Texture-Only) aligns closely with the distorted *Unicorn* point cloud (MOS: 4.591), indicating that the learning-based model concludes consistent with the FR metric. This verification underscores the significant impact of texture on geometry Gaussian noise.

*Interplay among the associations.* We've identified 6 association features in §3.3. To understand their contributions separately, To see how each feature contributes, we compared their cosine similarity to the final feature map before decoding [48]. By ranking these features based on similarity, we observe their influence on perceptual quality across distortion types and datasets, as depicted in Figure 4, we draw the following observations: (1) **Mixed Distortion in Colored Point Clouds**: The most important factor for quality here

Table 4: Performance comparison among the proposed metric with different variants on 4 datasets.

| Settings | SJTU-PCQA | | | WPC | | | BASICS | | | MJ-PCCD | | |
|---|---|---|---|---|---|---|---|---|---|---|---|---|
| | SRCC | PLCC | RMSE | SRCC | PLCC | RMSE | SRCC | PLCC | RMSE | SRCC | PLCC | RMSE |
| M3-Unity | **0.947** | **0.961** | 0.834 | **0.900** | **0.900** | **0.989** | **0.872** | **0.937** | **0.375** | **0.903** | 0.919 | 0.643 |
| Texture-Only | 0.942 | 0.956 | **0.675** | 0.895 | 0.894 | 1.021 | 0.855 | 0.905 | 0.457 | 0.874 | **0.927** | **0.413** |
| Geometry-Only | 0.888 | 0.915 | 0.948 | 0.644 | 0.670 | 1.692 | 0.837 | 0.905 | 0.677 | 0.818 | 0.860 | 0.561 |

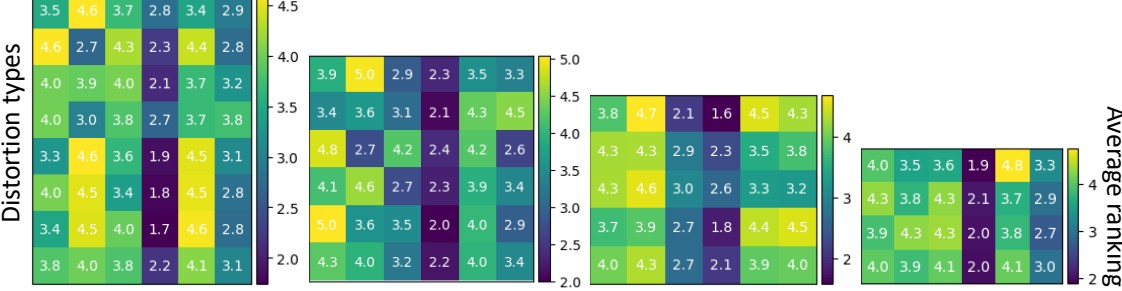

Figure 4: Visualization of the 6 associations' average rankings per distortion type across 4 datasets (tex2D_geo2D, tex3D_geo3D, tex2D_tex3D, tex2D_geo3D, geo2D_tex3D, geo2D_geo3D). **The result is computed in the same way as described in Sec §4** *Implementation details*. **Lower values indicate higher perceptual quality importance. The datasets in order from left to right are** SJTU-PCQA, WPC, BASICS**, and** MJ-PCCD**. The distortion types in order from top to down are as described in Sec §4** *datasets* **and overall ranking.**

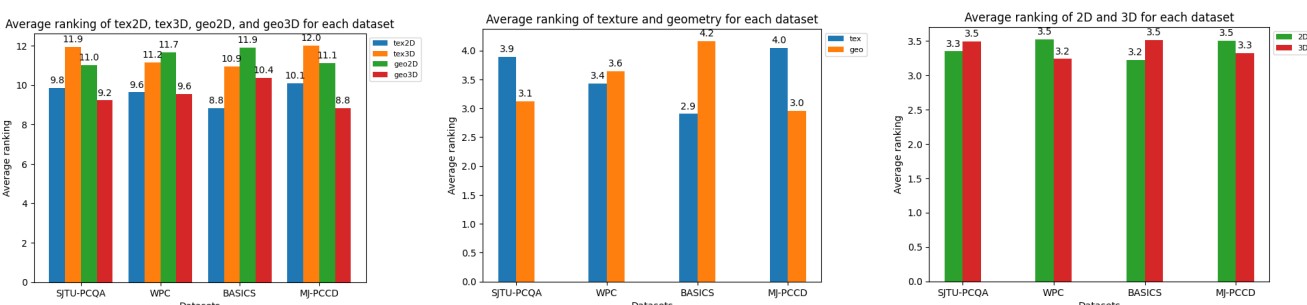

Figure 5: Average ranking grouped by different modality, attributes, and dimensionality. Each bar represents a ranking.

is the association between 2D texture and 3D geometry. Following closely is the association of geometry in both dimensionalities (SJTU-PCQA and MJ-PCCD) and texture in both dimensionalities (WPC and BASICS). The importance of the least crucial factor varies depending on the specific type of distortion. (2) **Compression**: VPCC and GPCC's quality is least influenced by 3D-related association. VPCC distorts 2D images due to its projection-based coding, while GPCC follows a geometry-based coding principle, with attribute coding relying on decoded geometry, making the correlations between 3D geometry and 3D texture less effective. (3) **Relative importance grouped by modalities, attributes and dimensionality**: The average ranking of them is shown in Figure 5, which is accumulated based on Figure 4. It shows that 2D texture and 3D geometry are most influential. Additionally, geometry distortion is more pronounced than texture for SJTU-PCQA and MJ-PCCD, since GPCC and JPEG Pleno MJ-PCCD can produce super dense/sparse stimuli and with uneven point distribution, and SJTU-PCQA has more types of geometric distortion. 3D distortion is more pronounced than 2D for WPC and MJ-PCCD datasets. Specifically, we computed the average ranking for GPCC and JPEG Pleno in MJ-PCCD, which are 3.38/3.68 and 2.7/4.3 for geometry/texture,

respectively, showing preference towards texture, which is in line with the conclusion from previous experiment[19].

## 6 CONCLUSIONS

In this paper, we introduce a novel no-reference framework designed for evaluating the quality of colored point clouds across multiple modalities and tasks. The self-attention mechanism is employed to fuse modality-related features, therefore enhancing the feature representations for quality assessment. Our framework enables a comprehensive measurement of the contributions stemming from both inter- and intra-associations, particularly concerning distinct distortion types relevant to perceptual quality assessment. In our investigations, we discovered that relying solely on 3D positional data may not suffice for accurately gauging geometric distortion, and the interplay between the attributes is crucial in understanding the overall distortion. We observed notable performance improvements by incorporating additional geometric information such as surface normals and association features. Furthermore, We draw conclusions about the prioritization of geometry/texture for point cloud quality assessment, providing valuable insights for bit allocation in point cloud compression and various high-level computer vision tasks.

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
