# OpenReview forum: "Deciphering Perceptual Quality in Colored Point Cloud: Prioritizing Geometry or Texture Distortion?"
_acmmm.org/ACMMM/2024/Conference — MM2024 Oral_

### Official Review · Reviewer_f1r4 · 2024-05-13

**Rating:** 5
**Confidence:** 4

**Summary:**

The paper introduces M3-Unity, a multi-task guided multi-modality no reference metric for evaluating the quality of colored point clouds. It utilizes four modalities across different attributes and dimensionalities to represent point clouds. The metric employs an attention mechanism to establish inter/intra associations among patches, yielding both local and global features to fit the human visual system's nonlinear properties. A multi-task decoder aids in distortion type classification, selecting the best combination of modalities for specific distortions. The framework enables in-depth analysis of the interplay between geometric and textural distortions, improving understanding in real-world applications. Experimental results on four benchmark datasets demonstrate significant performance gains over existing metrics.

**Strengths:**

M3-Unity outperforms state-of-the-art metrics across multiple datasets, indicating its effectiveness.
Provides insights into the interplay between geometric and textural distortions, advancing understanding in perceptual quality assessment.
Well-structured framework design enables efficient measurement of attribute contributions, aiding in real-world applications.
This paper is well written and I got inspiration from this paper and see certain further studies based on the insight and conclusion.

**Limitations:**

The multi-modal, multi-task approach may introduce complexity in implementation and understanding.Given that M3-Unity utilizes nearly all point cloud information, it's logical that it achieves top performance. It would be helpful if the authors also outline the complexities of M3-Unity variants, especially considering that although there's a slight performance drop, the complexity (image-only) could rival existing models.
In Sec5.2, more details are needed, one of them is that there are different distortion types in different datasets, how the authors deal with this situation when doing the cross-dataset validation.
The abbreviation HVS has no source, which I guess it is Human Vision System
In Fig1, the Point Cloud modality and Image Modality should be adjusted to adapt to the main context.
“The resulting feature h serves as the input to the decoder heads for final predictions, to be detailed as follows. “ At the end of Sec3.3 should be more clear to point out that the decoder should be introduced instead of h.
Why there is no performance on the BASICS dataset for GraphSim? Despite GraphSim's high complexity, it's essential to address the absence of reported figures for completeness.
The font in Fig3,5 is too small
Future work is missing in conclusion session

**Suitability:**

3

---

### Official Review · Reviewer_9MJ6 · 2024-05-20

**Rating:** 2
**Confidence:** 3

**Summary:**

This paper provides a point cloud quality assessment algorithm which achieves satisfying quality evaluation result by utilizing 4 types of modalities.

**Strengths:**

1. The multi-task decoder involving distortion type classification selects the best combination among 4 modalities based on the distortion type, aiding in the regression task.
2. An attention strategy is used to measure the impact of individual attributes and their combinations, providing insights into how these associations contribute particularly in relation to distortion type

**Limitations:**

1. For 2D modality, many geometric features have already been lost such as the curvature. Extracting geometric features from 2D modality seems meaningless.
2. The difference between this paper and the MM-PCQA is not very clear. The novelty of this paper need to be illustrated in detail.
3. In the ablation section, authors evaluate the impact of 2D projections and 3D point cloud respectively. However, the impact of 2D geometric, 2D texture, 3D geometric and 3D texture also need to be examined
4. In the introduction section, descriptions about the difference between subjective studies and objective studies are not very clear. It is quite confusing.
5. Some grammar mistakes exist in the paper, authors are suggested to check the whole paper carefully.

**Suitability:**

2

---

### Official Review · Reviewer_epai · 2024-05-23

**Rating:** 6
**Confidence:** 3

**Summary:**

The paper propose a multi-task guided multi-modality no reference metric for measuring the quality of colored point clouds, , which utilizes 4 types of modalities across different attributes and dimensionalities to represent point clouds. A multi-task decoder involving distortion type classification selects the best association among 4 modalities based on the specific distortion type, aiding the regression task and enabling the in-depth analysis of the interplay between geometrical and textural distortions.

**Strengths:**

1.	The paper is well-written and the techniques are sound.
2.	The proposed model achieves superior performance compared to previous NR-PCQA methods.
3.	The discussion about the interplay of geometry and texture features is good. It shows that 2D texture and 3D geometry are most influential for PCQA, which can inspire future researches.

**Limitations:**

1.	The figure 4 is somewhat confusing. The authors can divide the whole figure into four subfigures and annotate x/y axes of each subfigure with the corresponding caption.
2.	The overall format needs further improvement. For example, in Table 2, the positions of “train” and “test” could be adjusted.
3.	It is suggested to provide the parameter size for different modules.

**Suitability:**

3

---

### Meta-Review · Area_Chair_Ygjx · 2024-07-04

**Recommendation:** Accept (Oral)
**Confidence:** 5

**Metareview:**

The paper proposes a no-reference quality assessment method for point clouds, which is takes a multi-task guided multi-modality approach using 4 types of modalities across different attributes and dimensionalities to represent point clouds. The approach considered by the authors is interesting, also taking into account the attention mechanisms to understand the relationships among 2D and 3D patches. The experimental validation is quite complete, testing the method on several datasets and comparing with SOTA metrics. The authors also performed a good statistical analysis that added to the contributions of the paper.

The authors have carefully answered the concerns of the reviewers. I believe this paper is a good contribution to the area of point clouds and quality assessment.